# Towards Training Billion Parameter Graph Neural Networks for Atomic Simulations

**Anuroop Sriram**[†]**, Abhishek Das**[†]**, Brandon M. Wood**[‡→†]**, Siddharth Goyal**[†]**, C. Lawrence Zitnick**[†]

[†]Meta FAIR
[‡]National Energy Research Scientific Computing Center (NERSC)
{anuroops,abhshkdz,bmwood,sidgoyal,zitnick}@fb.com

## ABSTRACT

Recent progress in Graph Neural Networks (GNNs) for modeling atomic simulations has the potential to revolutionize catalyst discovery, which is a key step in making progress towards the energy breakthroughs needed to combat climate change. However, the GNNs that have proven most effective for this task are memory intensive as they model higher-order interactions in the graphs such as those between triplets or quadruplets of atoms, making it challenging to scale these models. In this paper, we introduce *Graph Parallelism*, a method to distribute input graphs across multiple GPUs, enabling us to train very large GNNs with hundreds of millions or billions of parameters. We empirically evaluate our method by scaling up the number of parameters of the recently proposed DimeNet++ and GemNet models by over an order of magnitude. On the large-scale Open Catalyst 2020 (OC20) dataset, these graph-parallelized models lead to relative improvements of 1) $15\%$ on the force MAE metric for the S2EF task and 2) $21\%$ on the AFbT metric for the IS2RS task, establishing new state-of-the-art results.

## 1 INTRODUCTION

Graph Neural Networks (GNNs) (Gori et al., 2005; Zhou et al., 2020) have emerged as the standard architecture of choice for modeling atomic systems, with a wide range of applications from protein structure prediction to catalyst discovery and drug design (Schütt et al., 2017b; Gilmer et al., 2017; Jørgensen et al., 2018; Zitnick et al., 2020; Schütt et al., 2017a; Xie & Grossman, 2018). These models operate on graph-structured inputs, where nodes of the graph represent atoms, and edges represent bonds or atomic neighbors. Despite their widespread success and the availability of large molecular datasets, training massive GNNs (with up to billions of parameters) is an important but under-explored area. Success of similarly large models in computer vision, natural language processing, and speech recognition (Shoeybi et al., 2020; Huang et al., 2019; Brown et al., 2020; Zhai et al., 2021) suggests that scaling up GNNs could yield significant performance gains.

Most previous approaches to scaling GNNs have focused on scaling small models (with up to a few million parameters) to massive graphs. These methods generally assume that we are working with a large, fixed graph, leading to the development of methods like neighborhood sampling (Jangda et al., 2021; Zheng et al., 2021) (Jia et al., 2020; Ma et al., 2019; Tripathy et al., 2020). These methods do not apply to atomic simulation datasets that contain millions of smaller graphs where it is necessary to consider the entire graph for prediction. Our focus is on the complementary problem of scaling to very large models for a dataset of many moderately-sized graphs ($\sim 1k$ nodes, $500k$ edges).

Another limitation of existing methods is that they focus on scaling simple GNN architectures such as Graph Convolutional Networks (GCNs) that only represent lower-order interactions *i.e.*, representations for nodes and edges, that are then updated by passing messages between neighboring nodes. In practice, the most successful GNNs used for atomic systems also model higher-order interactions between atoms, such as the interactions between triplets or quadruplets of atoms (Klicpera et al., 2020a; 2021; Liu et al., 2022). These interactions are necessary to capture the geometry of the underlying system, critical in making accurate predictions. Scaling such GNN architectures is challenging because even moderately-sized graphs can contain a large number of higher-order interactions. For example, a single graph with $1k$ nodes could contain several million triplets of atoms.

In this paper, we introduce *Graph Parallelism*, an approach to scale up such GNNs with higher-order interactions to billions of parameters, by splitting up the input graph across multiple GPUs.

We benchmark our approach by scaling up two recent GNN architectures – DimeNet++ (Klicpera et al., 2020a) and GemNet-T (Klicpera et al., 2021) – on the Open Catalyst (OC20) dataset (Chanussot* et al., 2021). The OC20 dataset, aimed at discovering new catalyst materials for renewable energy storage, consists of $134M$ training examples spanning a wide range of adsorbates and catalyst materials. A GNN that can accurately predict per-atom forces and system energies on OC20 has the potential to replace the computationally expensive quantum chemistry calculations based on Density Functional Theory (DFT) that are currently the bottleneck in computational catalysis. Our large-scale, graph-parallelized models lead to relative improvements of 1) $15\%$ for predicting forces on the force MAE metric (S2EF task), and 2) $21\%$ on the AFbT metric for predicting relaxed structures (IS2RS task), establishing new state-of-the-art results on this dataset.

## 2 GRAPH PARALLELISM

### 2.1 EXTENDED GRAPH NETS

Battaglia et al. (2018) introduced a framework called Graph Network (GN) that provides a general abstraction for many popular Graph Neural Networks (GNNs) operating on edge and node representations of graphs. We build on their work and define the Extended Graph Network (EGN) framework to include GNNs that also operate on higher order terms like triplets or quadruplets of nodes.

In the Graph Network (GN) framework, a graph is defined as a 3-tuple $\mathbf{G} = (\mathbf{u}, V, E)$, where $\mathbf{u}$ represents global attributes about the entire graph; $V = \{\boldsymbol{v}_i\}_{i=1:N^v}$ is the set of all nodes, with $\boldsymbol{v}_i$ representing the attributes of node $i$; and $E = \{(\mathbf{e}_k, r_k, s_k)\}_{k=1:N^e}$ is the set of all edges where $\mathbf{e}_k$ represents the attributes for the edge from node $s_k$ to $r_k$. A GNN then contains a series of GN blocks that iteratively operate on the input graph, updating the various representations.

In our *Extended Graph Network (EGN)* framework, a graph is defined as a 4-tuple $\mathbf{G} = (\mathbf{u}, V, E, T)$, where $\mathbf{u}$, $V$, and $E$ are defined as in the Graph Network, and $T = \{(\mathbf{t}_m, e_{m_1}, e_{m_2}, \ldots)\}$ is the set of higher-order interaction terms involving edges indexed by $m_1, m_2, \ldots$.

As a concrete example, consider an atomic system represented as a graph in this framework with the nodes representing the atoms and the edges representing atomic neighbors. The node attributes $v_i$ and edge attributes $e_k$ could represent the atom's atomic numbers and distances between atoms respectively. The higher order interactions could represent triplets of atoms, *i.e.*, pairs of neighboring edges with $\mathbf{t}_m$ representing the bond angle, which is the angle between edges that share a common node. Finally, the global attribute $\mathbf{u}$ can represent the energy of the system. For clarity of exposition, we will limit our discussion to triplets in the rest of the paper, but higher order interactions can be handled in a similar manner. We denote these triplets as $(\mathbf{t}_m, e_{m_1}, e_{m_2})$.

In the EGN framework, GNNs then contain a series of *EGN blocks* that iteratively update the graph. Each EGN block consists of several *update* and *aggregation* functions that are applied to transform an input graph $(\mathbf{u}, V, E, T)$ into an output graph $(\mathbf{u}', V', E', T')$. Each EGN block starts by updating the highest order interactions, which are then aggregated before updating the next highest order interaction. For instance, the triplet representations are first updated (using $\mathrm{TripletUpdate}$ function) and then aggregated ($\mathrm{TripletAggr}$) at each edge. These aggregated representations are then used to update the edge representation ($\mathrm{EdgeUpdate}$). Next, the edges going into a node are aggregated and used to update the node representations, and so on. This is illustrated in figure 1a.

Many GNNs can be cast in the EGN framework using appropriate update and aggregation functions.

### 2.2 GRAPH PARALLELISM FOR EXTENDED GRAPH NETS

Training large EGNs can be challenging even on moderately sized graphs because of the large memory footprint required in storing and updating the representation for each triplet, edge, and node. In many applications, the number of edges is one or two orders of magnitude larger than the number of nodes, while the number of triplets is one or two orders of magnitude larger than the number of edges. Thus, storing and updating the triplet representations is often the bottleneck in terms of GPU memory and compute. Many recent methods such as (Klicpera et al., 2020a; 2021) overcome this

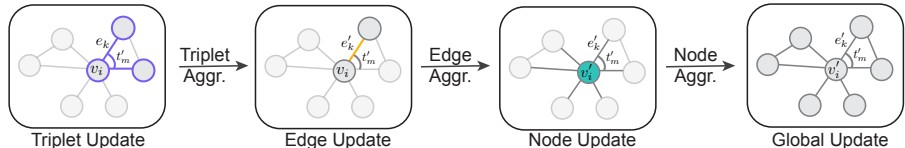

(a) Forward computation of an EGN block. First, each triplet representation is updated (TripletUpdate function), followed by aggregation of these updated representations (TripletAggr). Next, these aggregated values are used to update edge representations (EdgeUpdate), followed by edge aggregation (EdgeAggr), and finally node and global updates.

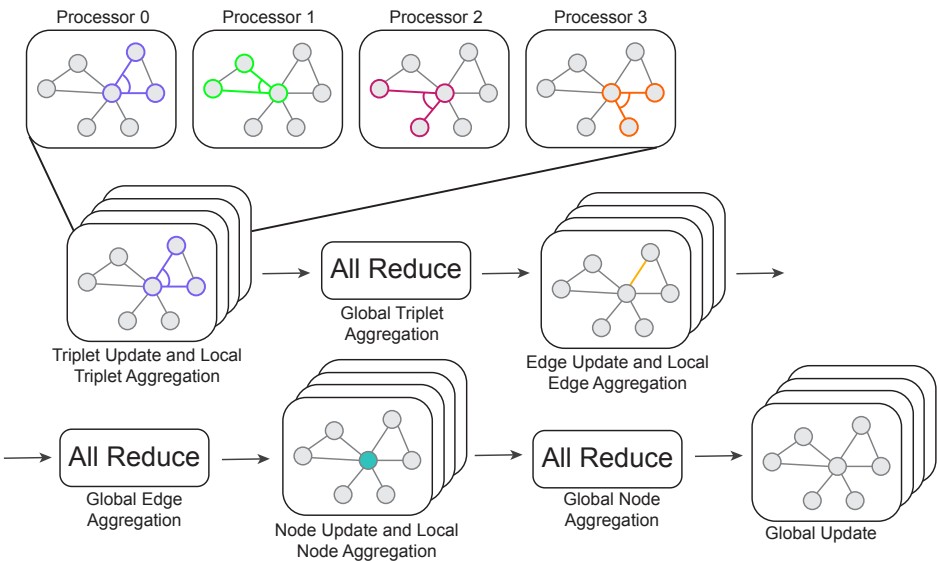

(b) Distributed computation of an EGN block. The graph is split up among the different GPUs such that processing unit $p$ contains its subset of triplets $T^{(p)}$ in memory, along with all edges $E$ and nodes $V$. The triplet attributes are updated in parallel, followed by a triplet aggregation to locally update the edge attributes. Next, an `allreduce` operation is performed to update all edge attributes globally. We continue this process to update the node and global attributes.

problem by using a very low-dimensional representation for the triplets. However, we found this to significantly reduce model capacity leading to underfitting for some applications with large training datasets. It is necessary to overcome these memory limitations for better generalization.

One way to avoid the memory limits is to distribute the computation across multiple GPUs. For a graph neural network, a natural choice to distribute the computation is by splitting the graph. The update functions in an EGN are easy to apply in parallel since they are applied independently for each triplet, edge, or node. It is substantially more challenging to apply the aggregations in parallel. To simplify parallel aggregation, we restrict ourselves to aggregation functions that are commutative and associative. This is not limiting in practice since most popular GNN architectures use sum or mean aggregations which can be implemented in this manner.

We will now describe the implementation of the distributed EGN block. Suppose we have access to $P$ processing units that we wish to split the graph computation over. Each unit $p$ is responsible for computing the updates to a subset of triplets, edges and nodes, that we denote by $T^{(p)}, E^{(p)}$, and $V^{(p)}$, respectively. At the beginning of computation, we split the graph so that the processing unit $p$ contains its subset of triplets $T^{(p)}$ in memory, along with the entire set of edges $E$ and nodes $V$. During the forward pass, we first update each set of triplets $T^{(p)}$ in parallel to obtain $T'^{(p)}$, followed by a local triplet aggregation. Next, an all-reduce operation is performed on these locally aggregated triplet representations to obtain globally aggregated triplet representations. These globally aggregated representations are then used to update edges in parallel, and the process is repeated for nodes and ultimately for the global graph level representation. Figure 1b shows this.

In this framework, the highest order interaction attributes are never communicated across processors. Therefore, the communication time is bound by the number of lower order node interactions. In our example, the triplet representations are not communicated, while the edge and node representations are communicated once per EGN block, making the total communication cost equal to $O(N_v D_v + N_e D_e)$, where $D_v$ and $D_e$ are the dimensions of node and edge representations. This makes it possible to work with a large number of triplets and large triplet representations. In section 3, we show concrete instantiations of this framework for two contemporary GNNs.

## 3    GRAPH PARALLELISM FOR ATOMIC SIMULATIONS

In this section, we present two concrete examples of using GNNs for the problem of predicting the energy and the forces for an atomic system, modeled as a graph whose nodes represent the atoms and whose edges represent the atoms' neighbors. The GNN takes such a graph as input and predicts the energy of the entire system as well as a 3D force vector on each atom. Two paradigms have been proposed in the literature, that we call *energy-centric* and *force-centric* approaches. These approaches differ in how they estimate forces and whether they are energy conserving, which is an important physical property. We begin by describing the components shared by both approaches, followed by one recent model in each paradigm.

### 3.1    INPUTS AND OUTPUTS

The inputs to the network are 3D positions $\mathbf{x}_i \in \mathbb{R}^3$ and atomic number $z_i$ for each atom $i \in \{1 \dots n\}$. The outputs are the per-atom forces $\mathbf{f}_i \in \mathbb{R}^3$ and the energy $E \in \mathbb{R}$ of the entire structure. The distance between atoms $i$ and $j$ is denoted as $d_{ij} = ||\mathbf{x}_i - \mathbf{x}_j||$. If edges $(i, j)$ and $(j, k)$ exist, then $(i, j, k)$ defines a *triplet* in the graph and we denote the angle between the edges as $\alpha_{kj,ji}$.

The input graph is constructed with each atom $t$ (*target*) as a node and the edges representing the atom's neighbors $s \in N_t$ where $N_t$ contains all atoms $s$ (*source*) within a distance $\delta$, which is treated as a hyperparameter. Each edge has a corresponding message $m_{st}$ that passes information from source atom $s$ to target atom $t$.

### 3.2    ESTIMATING FORCES AND ENERGY

As previously stated, there are two paradigms for estimating the energy and forces for an atomic system: *energy-centric* and *force-centric*. In energy-centric models, the model first computes the energy $E$ by applying a forward pass of the GNN: $E = \mathbf{GNN}(\mathbf{x}, \mathbf{z})$, where $\mathbf{x}$, and $\mathbf{z}$ represent the atomic positions and atomic numbers respectively. The forces are then computed as the negative gradient of the energy with respect to atomic positions by using backpropagation: $\mathbf{f} = -\nabla_{\mathbf{x}} E$.

In force-centric models, the energy and forces are both computed directly during the forward propagation: $E, \mathbf{F} = \mathbf{GNN}(\mathbf{x}, \mathbf{z})$ where $\mathbf{F}$ represents the matrix of all atomic forces. Force-centric models tend to be more efficient in terms of computation time and memory usage compared to the energy-centric models. However, energy-centric models guarantee that the forces are energy conserving which is an important physical property satisfied by atomic systems. In this work, we demonstrate the benefits of scaling up GNNs in both paradigms.

### 3.3    ENERGY-CENTRIC MODEL: DIMENET++

DimeNet++ (Klicpera et al., 2020a) is a recently proposed energy-centric model for atomic systems. In this model, the edges are represented by a feature representation $\mathbf{m}_{ji}$, which are iteratively updated using both directional information (via bond angles) as well as the interatomic distances. The edges are initially represented using a radial basis function (RBF) representation $\mathbf{e}_{RBF}^{(ji)}$ of their lengths $d_{ji}$. The triplets are represented using a spherical basis function (SBF) expansion $\mathbf{a}_{SBF}^{(kj,ji)}$ of the distances $d_{kj}$ as well as bond angles $\alpha_{(kj,ji)}$. In each block, the messages are updated as:

$$\mathbf{m}'_{ji} = f_{\text{update}}(\mathbf{m}_{ji}, \sum_{k \in \mathcal{N}_j \setminus \{i\}} f_{\text{int}}(\mathbf{m}_{kj}, \mathbf{e}_{RBF}^{(ji)}, \mathbf{a}_{SBF}^{(kj,ji)})) \tag{1}$$

where the interaction function $f_{int}$ corresponds to the $\mathrm{TripletUpdate}$ function, the summation corresponds to the $\mathrm{TripletAggr}$ function, and the update function $f_{update}$ to the $\mathrm{EdgeUpdate}$ function respectively. The interaction function consists of a Hadamard product of the embeddings, followed by a multi-layer perceptron. To speed up these computations, the embeddings are projected down to a smaller dimension before computing interactions, and later projected back up.

The updated messages are then fed as input to an output block that sums them up per atom $i$ to obtain a per-atom representation: $\mathbf{h}_i = \sum_j \mathbf{m}'_{ji}$ ($\mathrm{EdgeAggr}$ function). These are then transformed by another MLP to obtain the node representation $\boldsymbol{v}'_i$ ($\mathrm{NodeUpdate}$ function).

Thus, DimeNet++ is a case of EGN, and we closely follow the recipe from Sec. 2.2 to parallelize it.

## 3.4 FORCE-CENTRIC MODEL: GEMNET

GemNet (Klicpera et al., 2021) extends DimeNet++ in a number of ways, including the addition of quadruplets as well as a force-centric version. Here, we focus only on the force-centric GemNet-T model since it was recently shown to obtain state-of-the-art results on the OC20 dataset[1]. GemNet-T largely follows the structure of the DimeNet++ model, but includes some modifications.

First, GemNet-T uses a bilinear layer instead of the Hadamard product for the interaction function. This is made efficient by optimally choosing the order of operations in the bilinear function to minimize computation. Second, GemNet-T maintains an explicit embedding for each atom that is first updated by aggregating the directional embeddings involving that atom, similar to DimeNet++. Next, the updated atom embedding is used to update each of the edge embeddings. This creates a second edge update function, $\mathrm{EdgeUpdate}'$, that is run after the node embeddings are updated. Third, GemNet makes use of symmetric message passing, that is the messages $\mathbf{m}_{ji}$ and $\mathbf{m}_{ij}$ that are on the same edge, but in different directions, are coupled. In a parallel implementation, this step requires an additional all-reduce step since messages $\mathbf{m}_{ji}$ and $\mathbf{m}_{ij}$ could be on different processors.

Thus, GemNet-T largely follows the EGN framework, with a few minor deviations from the standard formulation. The distributed EGN implementation described in section 2.2 can be used for the GemNet-T as well, but with additional communication steps to account for the second edge update function and symmetric message passing.

## 4 EXPERIMENTS

In this section, we present the results of our scaling experiments on the Open Catalyst 2020 (OC20) dataset (Chanussot* et al., 2021). The OC20 dataset contains over 130 million atomic structures used to train models for predicting forces and energies during structure relaxations. We report results for three tasks: 1) Structure to Energy and Forces (S2EF) that involves predicting energy and forces for a given structure; 2) Initial Structure to Relaxed Energy (IS2RE) that involves predicting the relaxed energy for a given initial structure; and 3) Initial Structure to Relaxed Structure (IS2RS) which involves performing a structure relaxation using the predicted forces. DimeNet++ and GemNet-T are the current state-of-the-art energy-centric and force-centric models respectively.

## 4.1 EXPERIMENTAL SETUP

**DimeNet++-XL**. Our DimeNet++ model consists of $B = 4$ interaction blocks, with a hidden dimension of $H = 2048$, an output block dimension of $D = 1536$, and intermediate triplet dimension of $T = 256$. This model has about $240M$ parameters, which is over $20\times$ larger than the DimeNet++-large model used in (Chanussot* et al., 2021). We call this model *DimeNet++-XL*. The model was trained with the AdamW optimizer (Kingma & Ba, 2015; Loshchilov & Hutter, 2019) starting with an initial learning rate of $10^{-4}$, that was multiplied by $0.8$ whenever the validation error plateaus. The model was trained with an effective batch size of 128 on 256 Volta 32GB GPUs with a combination of data parallel and graph parallel training: each graph was split over 4 GPUs with data parallel training across groups of 4 GPUs.

---

[1] opencatalystproject.org/leaderboard.html

| Model | #Params | Training GPU-days | Energy MAE (eV)↓ | S2EF Test Force MAE (eV/A)↓ | Force Cos↑ | EFwT↑ |
|---|---|---|---|---|---|---|
| ForceNet-Large (Hu et al., 2021) | 34.8M | 194 | 2.2628 | 0.03115 | 0.5195 | 0.01% |
| DimeNet++-Large (Klicpera et al., 2020a) | 10.8M | 1600 | 31.5409 | 0.03132 | 0.5440 | 0.00% |
| SpinConv (Shuaibi et al., 2021) | 8.9M | 76 | 0.3363 | 0.02966 | 0.5391 | 0.45% |
| GemNet-T (Klicpera et al., 2021) | 31M | 47 | 0.2924 | 0.02422 | 0.6162 | 1.20% |
| GemNet-XL | 300M | 1962 | **0.2701** | **0.02040** | **0.6603** | **1.81%** |

Table 1: Experimental results on the S2EF task comparing our GemNet-XL to the top entries on the Open Catalyst leaderboard, showing metrics averaged across the 4 test datasets.

| Model | #Params | Training Dataset | IS2RS Test AFbT↑ | ADwT↑ | FbT↑ |
|---|---|---|---|---|---|
| SpinConv (Shuaibi et al., 2021) | 8.9M | S2EF-All | 16.67% | 53.62% | 0.05% |
| DimeNet++ (Klicpera et al., 2020a) | 1.8M | S2EF 20M + MD | 17.15% | 47.72% | 0.15% |
| DimeNet++-large (Klicpera et al., 2020a) | 10.8M | S2EF-All | 21.82% | 51.68% | 0.40% |
| GemNet-T (Klicpera et al., 2021) | 31M | S2EF-All | 27.60% | 58.68% | 0.70% |
| DimeNet++-XL | 240M | S2EF 20M + MD | **33.44%** | 59.21% | **1.25%** |
| GemNet-XL | 300M | S2EF-All | 30.82% | **62.65%** | 0.90% |

Table 2: Results on the IS2RS task comparing our models to the top entries on the Open Catalyst leaderboard, showing metrics averaged across the 4 test datasets. The DimeNet++ and DimeNet++-XL models were trained on the S2EF 20M + MD dataset, that contains additional molecular dynamics data and has been shown to be helpful for the IS2RS task (Chanussot* et al., 2021).

**GemNet-XL**. Our GemNet model consists of $B = 6$ interaction blocks, with an edge embedding size of $E = 1536$, triplet embedding size of $T = 384$ and embedding dimension of the bilinear layer of $B = 192$. We found that it was beneficial to use a small atom embedding size of $A = 128$, much smaller than the previous SOTA GemNet model[2]. This model has roughly 300M parameters, which is about $10\times$ larger than the previous SOTA model. However, since we reduced the atom embedding dimension and increased the edge and triplet dimensions, the total amount of compute and memory usage is significantly larger. We call this model *GemNet-XL*. We followed the same training procedure as with DimeNet++, except for a starting learning rate of $2 \times 10^{-4}$.

## 4.2 STRUCTURE TO ENERGY AND FORCES (S2EF)

The Structure to Energy and Forces task takes an atomic structure as input and predicts the energy of the entire structure and per-atom forces. The S2EF task has four metrics: the energy and force Mean Absolute Error (MAE), the Force Cosine Similarity, and the Energy and Forces within a Threshold (EFwT). EFwT indicates the percentage of energy and force predictions below a preset threshold.

Table 1 compares the top models on the Open Catalyst Project leaderboard[1] with the GemNet-XL model. GemNet-XL obtains a roughly $16\%$ lower force MAE and an $8\%$ lower energy MAE relative to the previous state-of-the-art. Further, GemNet-XL improves the EFwT metric more than $50\%$ relative to the previous best, although the value is still very small. These results indicate that model scaling is beneficial for the S2EF task.

## 4.3 INITIAL STRUCTURE TO RELAXED STRUCTURE (IS2RS)

The Initial Structure to Relaxed Structure (IS2RS) task involves taking an initial atomic structure and predicting the atomic structure that minimizes the energy. This is performed by iteratively predicting the forces on each atom and then using these forces to update the atomic positions. This process is repeated until convergence or 200 iterations. There are three metrics for this task: Average Distance within Threshold (ADwT), that measures the fraction of final atomic positions within a distance threshold of the ground truth; Forces below Threshold (FbT), which measures whether a true energy minimum was found (*i.e.*, forces are smaller than a preset threshold); and, the Average Forces below Threshold (AFbt), which averages the FbT over several thresholds.

Table 2 shows the results on the IS2RS task, comparing our models with the top few models on the Open Catalyst leaderboard. Both the DimeNet++-XL and GemNet-XL models outperform all

---

[2]`discuss.opencatalystproject.org/t/new-gemnet-dt-code-results-model-weights/102`

| Model | Approach | IS2RE Test | |
| | | Energy MAE (EV)↓ | EwT↑ |
|---|---|---|---|
| SpinConv (Shuaibi et al., 2021) | Relaxation | 0.4343 | 7.90% |
| GemNet-T (Klicpera et al., 2021) | Relaxation | 0.3997 | 9.86% |
| Noisy Nodes (Godwin et al., 2022) | Direct | 0.4728 | 6.50% |
| 3D-Graphormer (Ying et al., 2021) | Direct | 0.4722 | 6.10% |
| GemNet-XL | Relaxation | **0.3712** | **11.13%** |
| GemNet-XL-FT | Direct | 0.4623 | 5.60% |

Table 3: Results on the IS2RE task comparing our GemNet-XL to the top entries on the Open Catalyst leaderboard, showing metrics averaged across the 4 test datasets.

| #Blocks | Node Dim | Edge Dim | Trip Dim | Bil Dim | Params | #GP GPUs | #GP+DP GPUs |
|---|---|---|---|---|---|---|---|
| 3 | 1280 | 768 | 128 | 64 | 125M | 1 | 32 |
| 4 | 1536 | 1024 | 192 | 96 | 245M | 2 | 64 |
| 6 | 1792 | 1184 | 288 | 160 | 480M | 4 | 128 |
| 8 | 2320 | 1302 | 512 | 288 | 960M | 8 | 256 |

Table 4: Model hyperparameters for the scaling analysis. "#GP GPUs" denotes the number of GPUs over which the graph is distributed over for pure graph parallel training on a single node. "#GP+DP GPUs" denotes the total number of GPUs used to train with graph parallel training together with 32-way data parallel training.

existing models. The DimeNet++-XL model obtains a relative improvement of 53% on the AFbT metric, and more than triples the FbT metric compared to the DimeNet-large model. The GemNet-XL model obtains similar improvements compared to the smaller GemNet-T model. These results underscore the importance of model scaling for this task.

## 4.4    INITIAL STRUCTURE TO RELAXED ENERGY (IS2RE)

The Initial Structure to Relaxed Energy (IS2RE) task takes an initial atomic structure and attempts to predict the energy of the structure after it has been relaxed. Two approaches can be taken to address this problem, the direct and the relaxation approaches (Chanussot* et al., 2021). In the direct approach, we treat this task as a regression problem, and train a model to directly estimate the relaxed energy for a given atomic structure. The relaxation approach first estimates the relaxed structure (IS2RS task) after which the energy is estimated using the relaxed structure as input. Relaxation approaches typically outperform the direct approaches, though they are generally two orders of magnitude slower during inference time due to the need to estimate the relaxed structure.

Table 3 compares our GemNet-XL model to the top three models from the Open Catalyst Project leaderboard. There are two metrics for the IS2RE task: energy Mean Absolute Error (MAE) and the Energy within Threshold (EwT) which measures the percentage of time the predicted energy is within a threshold of the true energy. Table 3 shows that the GemNet-XL model obtains a roughly 8% lower energy MAE and a 12% higher EwT compared to the previous best, which is the smaller GemNet-T model, demonstrating the benefits of scaling up IS2RE models.

Since direct IS2RE models are very fast at inference time, there are applications where they are more useful than relaxation based approaches. It is possible to convert a trained S2EF model into a direct IS2RE model by fine-tuning it on the IS2RE training data. We finetune our GemNet-XL model in this manner for 5 epochs, starting with an initial learning rate of $3 \times 10^{-5}$ that is exponentially decayed by multiplying with 0.95 at the end of each epoch. The resuting model – GemNet-XL-FT – obtains a relative improvement of $\sim 2\%$ on energy MAE compared to 3D-Graphormer (Ying et al., 2021), the current state-of-the-art direct approach (Table 3).

## 4.5    SCALING ANALYSIS

Weak scaling studies the effect on the throughput when computation is scaled proportional to the number of processors. We study weak scaling efficiency in terms of scaling the model size propor-

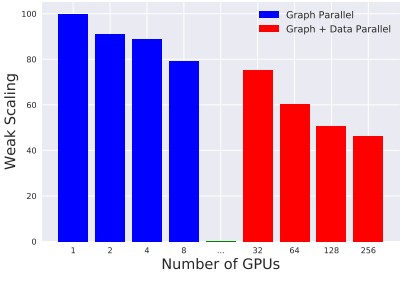

(a) Weak scaling efficiency.

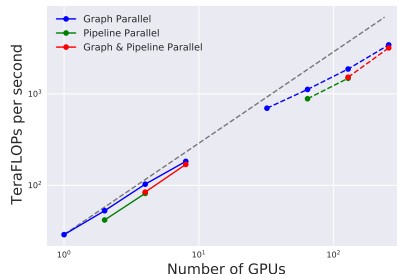

(b) Comparison of graph and pipeline parallel training.

Figure 2: **Left:** Weak scaling efficiency measured by scaling the model size proportional to the number of GPUs. **Right:** Comparing graph parallelism with pipeline parallelism. Solid blue, green and red curves show the scaling performance of graph, pipeline, and graph+pipeline parallel training. Dashed lines show the same with 32-way data parallel training. We do not show the results of training the largest models with pipeline parallelism as those runs ran out of GPU memory.

tional to the number of GPUs. For these experiments, we use 4 different GemNet-T models, ranging from 120M parameters to nearly 1B parameters (see Table 4).

We train increasingly larger models on multiple GPUs that do not fit on a single GPU. Figure 2a shows that we are able to obtain a scaling efficiency of roughly 79% with 8 GPUs for our largest model with nearly a billion parameters. This shows that our graph parallel training can be scaled up to billion parameter GemNet-T models while obtaining reasonably good scaling performance.

Figure 2a further shows the scaling efficiency for each model combining graph parallelism with 32-way data parallelism. Pure data parallel training with 32 GPUs obtains a scaling efficiency of 75% for the smallest model, showing the effect of network communication and load imbalance between the GPUs. Combining graph and data parallel training with 256 GPUs only reduces the scaling efficiency to 47% for the largest model compared to the 1 GPU case, suggesting the graph parallelism is promising for training extremely large models on hundreds of GPUs.

Finally, figure 2b shows the raw performance of running these models on V100 GPUs in terms of TeraFLOPs per second as a function of the number of GPUs. On a single GPU, the 120M parameter GemNet-T sustains 32 TeraFLOPs or roughly 25% of the theoretical peak FLOPS for a single GPU, and is thus a strong baseline. With 256 GPUs, the largest model sustains 3.5 PetaFLOPs.

Figure 2b compares graph and pipeline parallelism (Huang et al., 2019) showing that graph parallelism outperforms pipeline parallelism for the models that we consider. Since each graph in the training data contains different numbers of nodes, edges and triplets, load balancing across GPUs is difficult for pipeline parallelism. Graph parallelism is able to overcome this problem since the nodes, edges and triplets of a given batch are always distributed evenly across the GPUs, helping it outperform pipeline parallelism. It is possible, however, that pipeline parallelism might outperform graph parallelism for very deep GNNs since inter-GPU communication overhead for pipeline parallelism is independent of the number of blocks. Figure 2b also shows results with graph and pipeline parallelism combined, indicating that these methods are complementary to each other.

## 5 RELATED WORK

**GNNs for simluating atomic systems**  Many GNN based approaches have been proposed for the task of estimating atomic properties such as (Schütt et al., 2017b; Gilmer et al., 2017; Jørgensen et al., 2018; Schütt et al., 2017a; 2018; Xie & Grossman, 2018; Qiao et al., 2020; Klicpera et al., 2020b), where the atoms are represented by nodes and neighboring atoms are connected by edges. An early approach for force estimation was the SchNet model Schütt et al. (2017a), which computed forces using only the distance between atoms without the use of angular information. SchNet proposed the use of differentiable edge filters which enabled constructing energy-conserving models by estimating forces as the gradient of the energy. Subsequent work (Klicpera et al., 2020a;b; 2021; Liu et al., 2022) has extended on the SchNet model by adding bond angles and dihedral angles,

which has resulted in improved performance. These models make use of higher order interactions among nodes which make them highly compute and memory intensive. An alternate approach for estimating forces is to directly regress the forces as an output of the network. While this approach does not enforce energy conservation or rotational equivariance, recent work (Hu et al., 2021) has shown that such models can still accurately predict forces.

**Distributed GNN Training**. Research on distributed GNN training has focused on the regime of training small models on a single, large graph, for instance by sampling local neighborhoods around nodes to create mini-batches (Jangda et al., 2021; Zhang et al., 2020; Zhu et al., 2019). While these approaches can scale to very large graphs, they are not suitable for the task of modeling atomic systems where it is important to consider the entire graph for predicting the energy and forces.

An alternate line of work, that is more similar to ours, keeps the entire graph in memory by efficiently partitioning the graph among multiple nodes (Jia et al., 2020; Ma et al., 2019; Tripathy et al., 2020). These methods still operate in a single graph regime that is partitioned ahead of time. Thus the focus of that work is on finding efficient partitions of the graph, which is not applicable to our problem since we operate on millions of graphs. Further, these works do not train very large GNNs, or GNNs that operate on higher-order interactions (*e.g.* triplets), that are important for atomic systems.

**Model Parallelism**. methods focus on training large models that do not fit entirely on one GPU (even with a batch size of 1). GPipe (Huang et al., 2019) splits different sequences of layers into different processors, and splits each training mini-batch into micro-batches to avoid idle time per GPU. Megatron-LM (Shoeybi et al., 2020) splits the model breadth-wise, where Transformer layer weights are partitioned across multiple GPUs to distribute computation. We see model and graph parallelism as complementary approaches that can be combined to train even larger models.

# 6 DISCUSSION

We presented Graph Parallelism, a new method for training large GNNs for modeling atomic simulations, where modeling higher-order interactions between atoms (triplets / quadruplets) is critical. We showed that training larger DimeNet++ and GemNet-T models can yield significant improvements on the OC20 dataset. Although we demonstrated graph parallelism for just two GNNs, it is broadly applicable to a host of message-passing GNNs, including equivariant networks, that can be cast in the GN / EGN framework (Sec. 2.1) by appropriately picking update and aggregate functions.

Further, it is possible to combine graph parallelism with model parallel methods such as GPipe (Huang et al., 2019) to train even larger models, which could yield further improvements, as briefly explored in Sec 4.5. For force-centric GNNs, it should be possible to use graph parallel for 'breadth-wise' scaling *i.e.*, to split higher-order computations (*e.g.* triplets) across GPUs, and GPipe for 'depth-wise' scaling *i.e.*, to scale to larger number of message-passing blocks, sequentially split across GPUs. For energy-centric GNNs *e.g.*, DimeNet++, this combination is less obvious since these models require an additional backward pass through the network to compute forces as gradients of energy with respect to atomic positions. Energy-centric GNNs are common for atomic systems because they enforce the physical constraint of energy conservation. As we demonstrate with our DimeNet++-XL experiments, graph parallelism is applicable to energy-centric GNNs.

We see scaling to large model sizes as a necessary (but not sufficient) step to effectively model atomic simulations from large, diverse datasets of adsorbates and catalysts. Further progress may require marrying large scale with ways to better capture 3D geometry and physical priors.

The carbon emissions from training large deep learning models are non-trivial and the work we have presented here is no exception (Strubell et al., 2019; Schwartz et al., 2019). We estimate that training our GemNet-XL model with Tesla v100 32 GB GPUs on cloud resources in the US ranges from 3490 - 8052 kg of $CO_2$ eq. (Lacoste et al., 2019). In the worst case, the emissions are roughly equivalent to 16 round trip flights from Los Angeles to New York. Assuming that the training time is fixed, emissions largely depend on the carbon intensity of the energy generation in a given region and the percentage of emissions offset with other investments by the resource provider. When choosing compute resources we recommend evaluating the stated carbon offset commitments and if possible, consider running experiments in regions that have more sustainable energy generation. The compute resources we utilized for this paper were committed to be $100\%$ offset.

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

# A APPENDIX

## A.1 ADDITIONAL RESULTS

Tables 5, 6 and 7 show results from each of the four test sets for the S2EF, IS2RE and IS2RS tasks respectively. DimeNet++-XL and GemNet-XL achieve the best results for each test set along each metric.

| Model | S2EF Test | | | |
| --- | --- | --- | --- | --- |
| | Energy MAE (EV)↓ | Force Cos↑ | Force MAE (EV/A)↓ | EFwT↑ |
| **ID** | | | | |
| GemNet-T | 0.2257 | 0.637 | 0.02099 | 2.4% |
| SpinConv | 0.2612 | 0.5479 | 0.02689 | 0.82% |
| ForceNet-Large | 2.0674 | 0.533 | 0.02782 | 0.02% |
| DimeNet++-large | 29.3333 | 0.5633 | 0.02807 | 0% |
| GemNet-XL | **0.2120** | **0.6759** | **0.0181** | **3.30%** |
| **OOD Ads** | | | | |
| GemNet-T | 0.2099 | 0.6242 | 0.02186 | 1.15% |
| SpinConv | 0.2752 | 0.5345 | 0.02769 | 0.38% |
| ForceNet-Large | 2.4188 | 0.5212 | 0.02834 | 0.01% |
| DimeNet++-large | 30.0338 | 0.5503 | 0.02896 | 0% |
| GemNet-XL | **0.1980** | **0.6642** | **0.0186** | **1.62%** |
| **OOD Cat** | | | | |
| GemNet-T | 0.3403 | 0.5813 | 0.02445 | 0.93% |
| SpinConv | 0.3501 | 0.5187 | 0.02849 | 0.46% |
| ForceNet-Large | 2.0203 | 0.4936 | 0.03089 | 0.01% |
| DimeNet++-Large | 30.0437 | 0.5109 | 0.0312 | 0% |
| GemNet-XL | **0.3083** | **0.6306** | **0.0206** | **1.72%** |
| **OOD Both** | | | | |
| GemNet-T | 0.3937 | 0.6221 | 0.02956 | 0.3% |
| SpinConv | 0.4585 | 0.5554 | 0.03556 | 0.14% |
| Forcenet-Large | 2.5447 | 0.5302 | 0.03754 | 0% |
| DimeNet++-Large | 36.7529 | 0.5517 | 0.03705 | 0% |
| GemNet-XL | **0.362** | **0.6704** | **0.0245** | **0.61%** |

Table 5: Full set of results on the S2EF task comparing our GemNet-XL to the top three entries on the Open Catalyst leaderboard, showing metrics from each test set.

| Model | Approach | IS2RE Test | |
| --- | --- | --- | --- |
| | | Energy MAE (EV)↓ | EwT↑ |
| **ID** | | | |
| GemNet-T | Relaxation | 0.3901 | 12.37% |
| SpinConv | Relaxation | 0.4207 | 9.4% |
| Noisy Nodes | Direct | 0.4776 | 5.71% |
| GemNet-XL | Relaxation | **0.3764** | **13.25%** |
| GemNet-XL-FT | Direct | 0.4194 | 7.52% |
| **OOD Ads** | | | |
| GemNet-T | Relaxation | 0.3907 | 9.11% |
| SpinConv | Relaxation | 0.4381 | 7.47% |
| Noisy Nodes | Direct | 0.5646 | 3.49% |
| GemNet-XL | Relaxation | **0.3677** | **10.00%** |
| GemNet-XL-FT | Direct | 0.5258 | 3.95% |
| **OOD Cat** | | | |
| GemNet-T | Relaxation | 0.4339 | 10.09% |
| SpinConv | Relaxation | 0.4579 | 8.16% |
| Noisy Nodes | Direct | 0.4932 | 5.02% |
| GemNet-XL | Relaxation | **0.4022** | **11.61%** |
| GemNet-XL-FT | Direct | 0.4373 | 6.76% |
| **OOD Both** | | | |
| GemNet-T | Relaxation | 0.3843 | 7.87% |
| SpinConv | Relaxation | 0.4203 | 6.56% |
| Noisy Nodes | Direct | 0.5042 | 3.82% |
| GemNet-XL | Relaxation | **0.3383** | **9.65%** |
| GemNet-XL-FT | Direct | 0.4665 | 4.19% |

Table 6: Experimental results on the IS2RE task comparing our GemNet-XL to the top three entries on the Open Catalyst leaderboard, showing metrics from each test set.

| Model | Training Dataset | IS2RS Test | | |
|---|---|---|---|---|
| | | AFbT↑ | ADwT↑ | FbT↑ |
| **ID** | | | | |
| GemNet-T | S2EF-ALL | 33.75% | 59.18% | 2.0% |
| DimeNet++-large | S2EF-ALL | 25.65% | 52.45% | 1.0% |
| SpinConv | S2EF-ALL | 21.10% | 53.68% | 0.2% |
| DimeNet++ | S2EF 20M + MD | 21.08% | 48.6% | 0.2% |
| DimeNet++-XL | S2EF 20M + MD | **40.00%** | 59.90% | **2.4%** |
| GemNet-XL | S2EF-ALL | 34.61 | **62.73%** | **2.4%** |
| **OOD Ads** | | | | |
| GemNet-T | S2EF-ALL | 26.84% | 54.59% | 0.2% |
| DimeNet++-large | S2EF-ALL | 20.73% | 48.47% | 0.4% |
| SpinConv | S2EF-ALL | 15.70% | 48.87% | 0.0% |
| DimeNet++- | S2EF 20M + MD | 17.05% | 42.98% | 0.0% |
| DimeNet++-XL | S2EF 20M + MD | **36.01%** | 55.68% | **1.6%** |
| GemNet-XL | S2EF-ALL | 30.32% | **58.57%** | 0.6% |
| **OOD Cat** | | | | |
| GemNet-T | S2EF-ALL | 24.69% | 58.71% | 0.4% |
| DimeNet++-large | S2EF-ALL | 20.24% | 50.99% | 0% |
| SpinConv | S2EF-ALL | 15.86% | 53.92% | 0% |
| DimeNet++- | S2EF 20M + MD | 16.43% | 48.19% | 0% |
| DimeNet++-XL | S2EF 20M + MD | **29.62%** | 58.43% | **0.6%** |
| GemNet-XL | S2EF-ALL | 29.33% | **62.60%** | 0.2% |
| **OOD Both** | | | | |
| GemNet-T | S2EF-ALL | 25.11% | 62.23% | 0.2% |
| DimeNet++-large | S2EF-ALL | 20.67% | 54.82% | 0.2% |
| SpinConv | S2EF-ALL | 14.01% | 58.03% | 0% |
| DimeNet++- | S2EF 20M + MD | 14.02% | 51.09% | **0.4%** |
| DimeNet++-XL | S2EF 20M + MD | 28.14% | 62.85% | **0.4%** |
| GemNet-XL | S2EF-ALL | **29.02%** | **66.72%** | **0.4%** |

Table 7: Experimental results on the IS2RS task comparing our models to the top four entries on the Open Catalyst leaderboard, showing metrics for each test dataset. The DimeNet++ and DimeNet++-XL models were trained on the S2EF 20M + MD dataset, that contains additional molecular dynamics data, which has been shown to be helpful for the IS2RS task.

