# OpenReview forum: "Towards Training Billion Parameter Graph Neural Networks for Atomic Simulations"
_ICLR.cc/2022/Conference — ICLR 2022 Poster_

### Official Review · Reviewer_ZcRk · 2021-11-02

**Correctness:** 3
**Technical Novelty And Significance:** 3
**Empirical Novelty And Significance:** 3
**Recommendation:** 5
**Confidence:** 1

**Main Review:**

The paper is based on the idea that most of the computation costs to extended graphs come from triplets. The authors proposed a way to parallelize the computation of triplets in a distributed way. The author also discussed two models that fit into this framework and showed their increase in performance due to the larger parameter count enabled by their parallel framework.

The paper is well written and easy to follow. A minor part I am a bit confused about is that in the sentence “In many applications, the number of edges is one or two orders of magnitude larger than the number of nodes, while the number of triplets is one or two orders of magnitude larger still”, do you mean the triplets are one or two orders larger than the node or the edges?

In the experiment section, network structures and hyperparameters are given and the experiment should be reproducible. The results support the authors' claim that larger graph networks have better performance.

I do not know much about this field.

**Summary Of The Paper:**

The paper is based on the idea that most of the computation costs to extended graphs come from triplets. The authors proposed a way to parallelize the computation of triplets in a distributed way. The author also discussed two models that fit into this framework and showed their increase in performance due to the larger parameter count enabled by their parallel framework.

**Summary Of The Review:**

The paper is based on the idea that most of the computation costs to extended graphs come from triplets. The authors proposed a way to parallelize the computation of triplets in a distributed way. The author also discussed two models that fit into this framework and showed their increase in performance due to the larger parameter count enabled by their parallel framework.

The paper is well written and easy to follow. A minor part I am a bit confused about is that in the sentence “In many applications, the number of edges is one or two orders of magnitude larger than the number of nodes, while the number of triplets is one or two orders of magnitude larger still”, do you mean the triplets are one or two orders larger than the node or the edges?

In the experiment section, network structures and hyperparameters are given and the experiment should be reproducible. The results support the authors' claim that larger graph networks have better performance.

I do not know much about this field.

---

> ### Author Response · Authors · 2021-11-22
> **Response to ZcRk**
>
> Thanks for the review!
>
> > A minor part I am a bit confused about is that in the sentence “In many applications, the number of edges is one or two orders of magnitude larger than the number of nodes, while the number of triplets is one or two orders of magnitude larger still”, do you mean the triplets are one or two orders larger than the node or the edges?
>
> The number of triplets is 1-2 orders of magnitude larger than the number of edges (so, 1-4 orders of magnitude larger than the number of nodes). This makes the number of triplets the primary bottleneck for these models. We have clarified this in the paper.

---

### Official Review · Reviewer_N2CL · 2021-11-02

**Correctness:** 3
**Technical Novelty And Significance:** 2
**Empirical Novelty And Significance:** 3
**Recommendation:** 6
**Confidence:** 3

**Main Review:**

Strength:
 * Good empirical results: OC20 is an open, comprehensive benchmark and the method achieved decent improvement over strong baselines in the leaderboard.

Weakness and questions:
 * Since this paper is about distributed training, it should be compared with other distributed training methods, including data and model parallelism. Does the new method run faster than data/model parallelism? For data parallelism, one could consider splitting the batch to multiple GPUs. The paper states that model parallelism can be combined with the proposed approach, but I think this paper should compare with standard model parallelism, such as pytorch DDP. If memory is the issue, combining data and model parallelism should be a reasonable baseline.
 * I am not so sure about the novelty of the distributed training approach here. It simply splits the graph to different GPUs and apply "All Reduce" after each layer of computation. This strategy doesn't seem to be efficient. Using 8 GPUs, the method can only reduce run time by 50%.

**Summary Of The Paper:**

This paper proposes a distributed training method for large graph neural networks (GNN) up to billion parameters. The method first distributes the triplet update operations to multiple GPUs and aggregate the updated vectors by global synchronization. It then distributes the edge update operations to GPUs and aggregate the edge vectors (another global synchronization). Finally, it applies node update in parallel and global node aggregation at the end. The method is applied to GemNet and DimeNet up to 1.12 billion parameters and achieved state-of-the-art results on OC20 benchmark.

**Summary Of The Review:**

Despite the good empirical results, it is not clear what is the benefit of the proposed approach over standard distributed training techniques (data/model parallelism). Since this paper proposes a new distributed training algorithm for graphs, it is necessary to compare the speedup of the proposed method against standard distributed training techniques.

---

> ### Author Response · Authors · 2021-11-22
> **Response to N2CL**
>
> Thanks for the review!
>
> > Since this paper is about distributed training, it should be compared with other distributed training methods, including data and model parallelism.
>
> > it is not clear what is the benefit of the proposed approach over standard distributed training techniques (data/model parallelism).
>
> Thanks for the feedback!
>
> Please see our common response comparing graph parallelism with data and model parallelism.
>
> To reiterate, all three approaches are complementary. Data parallel enables scaling to larger minibatch sizes with more processors, model parallel approaches such as GPipe (Huang et al., NeurIPS 2019) enable ‘depth-wise’ scaling i.e. to increase the number of message-passing blocks beyond what fits on a single processor, and our graph parallel enables ‘breadth-wise’ scaling i.e. to fit higher-order triplet / quadruplet interactions beyond what fits on a single processor.
>
> All our experiments make use of data parallel and graph parallel.
>
> We have updated Sec. 6 in our draft to clarify this.
>
> > This strategy doesn't seem to be efficient. Using 8 GPUs, the method can only reduce run time by 50%.
>
> There may have been a misunderstanding in interpreting the ~50% of linear scaling efficiency shown in Fig. 2b. To clarify, in Fig. 2b, we scale to 8x the model size on 8 GPUs achieving ~50% of linear scaling. For reference, if the throughput (time per batch) with 8x the model size on 8 GPUs was about the same as 1x the model size on 1 GPU, efficiency would be 100%. Going from 500M parameters (on 4 GPUs) to 1.12B parameters (on 8 GPUs), scaling efficiency only decreases from ~54% to ~50%, suggesting that graph parallelism is particularly useful in this large model size regime.
>
> To the best of our knowledge, graph parallelism is the only approach that enables scaling graph networks to large model sizes that wouldn’t otherwise fit on a single GPU. Please see our common response comparing graph parallelism with data and model parallelism for more details.

---

> > ### Comment · Reviewer_N2CL · 2021-12-02
> > **Thank you for your response**
> >
> > I have read your response and I would like to raise my score to 6 (weak accept). The new results suggest that data/model parallelism is complementary to this approach.

---

### Official Review · Reviewer_BwW9 · 2021-11-05

**Correctness:** 3
**Technical Novelty And Significance:** 3
**Empirical Novelty And Significance:** 3
**Recommendation:** 5
**Confidence:** 3

**Main Review:**

The approach is well-motivated and well-explained. I am familiar with the DimeNet architecture but not very familiar with the recent literature on distributed GNN training, so it is hard for me to judge the novelty of the proposed parallel approach. The results on the OpenCatalyst benchmark are compelling.

Among graph neural networks for molecular structures, the DimeNet++ architecture is specifically memory and compute intensive (and as such naturally stands to specifically benefit from the proposed approach). It would be great if the authors could include a discussion of the applicability of their approach in the context of other graph neural networks for molecular structure, such as equivariant message passing in Jing et al. (ICLR, 2021) or Schuett et al. (ICML, 2021).

The paper does not put the scaling results in Figure 2 in context with other approaches for parallel training.

**Summary Of The Paper:**

The authors present an approach to train graph neural networks with many parameters across multiple GPUs. The approach is specifically demonstrated for graphs representing molecular structure and two graph neural networks of similar flavor that have previously been presented to learn from such structures. Higher-order interaction terms make these networks very compute intensive. Leveraging their parallel approach to scale up the existing GNNs, the authors demonstrate compelling results on the OpenCatalyst benchmark.

**Summary Of The Review:**

I did not identify technical concerns with respect to the proposed method. I will defer to other reviewers regarding the technical novelty of the approach. The paper could be made stronger by explicitly discussing how this approach is applicable to other GNN architectures that have been designed for 3D molecular structure.

---

> ### Author Response · Authors · 2021-11-22
> **Response to BwW9**
>
> > It would be great if the authors could include a discussion of the applicability of their approach in the context of other graph neural networks for molecular structure, such as equivariant message passing in Jing et al. (ICLR, 2021) or Schuett et al. (ICML, 2021).
>
> > The paper could be made stronger by explicitly discussing how this approach is applicable to other GNN architectures that have been designed for 3D molecular structure.
>
> The Extended Graph Network (EGN) framework introduced in Sec. 2 in the paper builds upon the previously proposed Graph Network (GN) framework (Battaglia et al., 2018). Our graph parallel approach can be applied to any GNN that can be cast in the GN / EGN framework by an appropriate choice of update and aggregate functions.
>
> For example, the model proposed in Jing et al. (ICLR 2021) uses geometric representations for nodes and edges that are updated through message passing. The update functions involve the use of Geometric Vector Perceptrons (GVPs) that preserve equivariance. This model can be cast in the GN framework by using an EdgeUpdate function that includes GVPs within it.
>
> Similarly, Schuett et al. (ICML, 2021) present another equivariant GNN model called PAINN, that is composed of a series of Message and Update blocks. By viewing the computation within the message block as the EdgeUpdate function, the message passing step as the EdgeAggregate function and the Update block as the NodeUpdate function, we can see that this model can also be implemented as a GN.
>
> We have added a discussion on this to Sec. 6 in the paper.
>
> > The paper does not put the scaling results in Figure 2 in context with other approaches for parallel training.
>
> Please see our common response comparing graph parallelism with data and model parallelism.
>
> Fig. 2 demonstrates scaling behavior for graph parallelism in two settings -- when increasing the batch size (Fig. 2a) or the model size (Fig. 2b) linearly with no. of GPUs.
>
> In Fig. 2a, we see that using graph parallelism to scale to 8x the batch size on 8 GPUs achieves ~46% of linear scaling. For reference, if the throughput (time per batch) with 8x the batch size on 8 GPUs was about the same as 1x the batch size on 1 GPU, efficiency would be 100%. In this case, since the model fits on a single GPU, a comparison with data parallelism is possible, and we expect data parallelism to have better scaling efficiency for increasing batch size than graph parallelism because of significantly fewer quantities being communicated across GPUs than graph parallelism.
>
> In Fig. 2b, we scale to 8x the model size on 8 GPUs achieving ~50% of linear scaling. Here, a comparison with only data parallelism is not even possible since this model does not fit on 1 GPU. Going from 500M parameters (on 4 GPUs) to 1.12B parameters (on 8 GPUs), scaling efficiency only decreases from ~54% to ~50%, suggesting that graph parallelism is particularly useful in this large model size regime. To the best of our knowledge, graph parallelism is the only approach that enables scaling graph networks to large model sizes that wouldn’t otherwise fit on a single GPU.
>
> We have now added Fig. 2c as well that explicitly demonstrates scaling behavior just with graph parallel (on up to 8 GPUs) and when combining graph and data parallelism (on up to 256 GPUs).

---

### Official Review · Reviewer_4Syd · 2021-11-05

**Correctness:** 3
**Technical Novelty And Significance:** 3
**Empirical Novelty And Significance:** 4
**Recommendation:** 8
**Confidence:** 2

**Main Review:**

Strengths:

Overall the paper is well structured

Methods that enable effective training of large scale GNNs on datasets containing many graphs could be very impactful

The large GNN models are benchmarked on a set of tasks that are important in the real world (catalyst design), and show quite large improvements in performance compared to the smaller baselines

Weaknesses:

Not sure how reproducible this work is without code

Other comments/questions:

Some additional information on the computational cost of Gemnet-XL and Dimenet++-XL models in the 3 tasks would be useful. Eg wall time, gpu hours, etc. Although methods to train very large GNN models can be very impactful, if the computational cost is prohibitive for a lot of people then that could limit its impactfulness.


**Summary Of The Paper:**

This paper proposes a method to train large scale graph neural networks containing up to billions of parameters, called Graph Parallelism. The method is used to train large scale versions of the DimeNet++ and GemNet models containing 10-20x more parameters that the vanilla versions. These large GNN models are evaluated on a set of tasks from the Open Catalyst 2020 (OC20) benchmark and show improved performance compared to the smaller baselines.

**Summary Of The Review:**

Overall, I vote for acceptance. The paper proposes a method to train large scale graph neural networks and shows that very large GNN models can have quite large improvements in the Open Catalyst 2020 (OC20) benchmark

---

> ### Author Response · Authors · 2021-11-22
> **Response to 4Syd**
>
> Thanks for the review!
>
> > Some additional information on the computational cost of Gemnet-XL and Dimenet++-XL models in the 3 tasks would be useful.
>
> | Model                                   | # Params | Training GPU-days | Test Force Cos |
> |-----------------------------------------|----------|-------------------|----------------|
> | GemNet-T (Klicpera et al., 2021)       | 3.1M     | 47                | 0.6162      |
> | SpinConv (Shuaibi et al., 2021)         | 8.9M     | 76                | 0.5391        |
> | ForceNet-Large (Hu et al., 2021)        | 34.8M    | 194               | 0.5195        |
> | DimeNet++-Large (Klicpera et al., 2020) | 10.8M    | 1600              | 0.544        |
> | DimeNet++-XL (our work)                 | 240M     |         4546          |      0.575         |
> | GemNet-XL (our work)                    | 300M     |       1962            | 0.6603         |
>
> Thanks for the feedback! We have added computational cost in GPU-days to Table 1 in the paper.
>
> Our general observation has been that the larger graph-parallelized models converge faster than smaller models in terms of the number of epochs through the dataset.
>
> And while our focus in this work is to scale up model capacity to achieve the best scaling behavior across GPUs and overall state-of-the-art results, lowering computational cost is also of interest to us, something we aim to focus on in future work.
>
> > Not sure how reproducible this work is without code
>
> We will make sure to open-source our code for others to reproduce our results and build on.

---

> > ### Comment · Reviewer_4Syd · 2021-12-03
> > **Response**
> >
> > Thanks to the authors for the response. After reading all the other reviews and responses, I leave my initial score unchanged, but will reduce my confidence from 3 to 2, since I think there are some areas in the paper brought up by others that I did not fully appreciate before.

---

### Author Response · Authors · 2021-11-22
**Response to all reviewers**

We thank the reviewers for their feedback! We are encouraged that the reviewers find our results on the Open Catalyst benchmark strong (4Syd, BwW9, N2CL) and the paper well-written and easy to follow (ZcRk, BwW9, 4Syd). We address specific concerns below and have updated the draft to incorporate feedback.

* How does graph parallel compare to data parallelism and model parallelism?

Data Parallel training is applicable when the model is small enough to fit on a single processor (with a batch size of 1 or more per processor). The training minibatch can then be distributed across processors and scales linearly with the number of processors.

Model Parallelism is employed when the model does not fit entirely on one processor (even with a batch size of 1). As discussed in Section 5 in the paper, particularly relevant model parallelism approaches are GPipe (Huang et al., 2019) and Megatron-LM (Shoeybi et al., 2020). GPipe splits different sequences of layers into different processors, and splits each training minibatch into microbatches to avoid idle time per processor. Megatron-LM splits Transformer models ‘breadth-wise’, where each layer’s weights are partitioned across multiple GPUs.

As such, graph parallelism is complementary to both data parallelism and GPipe approaches. This is demonstrated in our paper, as we used data parallelism to scale up minibatch sizes, and  graph parallelism was used to scale model sizes beyond those that could fit on a single GPU. We have added Fig. 2c in the paper to demonstrate scaling behavior when combining graph and data parallelism.

We have not yet explored the use of GPipe with graph parallelism, but for force-centric GNNs, it should be possible to combine the two to scale model capacity further -- graph parallel for ‘breadth-wise’ scaling i.e. to split higher-order triplet / quadruplet computations across GPUs and GPipe for ‘depth-wise’ scaling i.e. to scale to larger number of message-passing blocks, sequentially split across GPUs.

For energy-centric GNNs e.g. DimeNet++, it is not obvious how GPipe would be implemented, since they require an additional backward pass through the network to compute forces as gradients of energy with respect to atomic positions. Energy-centric GNNs are common for atomic systems because they enforce the physical constraint of energy conservation. As we demonstrate with our DimeNet++-XL experiments, graph parallelism is applicable to energy-centric GNNs.

Similar to graph parallelism, Megatron-LM performs ‘breadth-wise’ scaling, but while graph parallelism partitions higher-order _inputs_ (e.g. triplets / quadruplets) across GPUs, Megatron-LM partitions Transformer layer _weights_ across GPUs.

---

### Decision · Program_Chairs · 2022-01-20

**Decision:**

Accept (Poster)

**Comment:**

The reviewers were split about this paper: on one hand they appreciated the clarity and the experimental improvments in the paper, on the other they were concerned about the novelty of the work. After going through it and the discussion I have decided to vote to accept this paper for the following reasons: (a) the potential impact of the work, (b) the simplicity of the idea, and (c) promise of release of open source code. I think these things make the paper a strong contribution to ICLR. The only thing I would like to see added, apart from the suggestions detailed by the reviewers, is a small discussion on the carbon footprint of training such largescale graph networks. The authors motivated the work by saying it could have a beneficial impact for modelling energy which is needed to combat climate change. However, we know from recent results that such large scale models also have a non-trivial emission footprint. So I'd like to see the authors specifically calculate the carbon footprints of the models they trained. There are tools to help with this such as: https://mlco2.github.io/impact/  With this addition I think this paper will not only make a large impact on graph network training but also start a discussion of how to responsibly decide training, taking environmental impact into account.